# Hybrid Weight Representation: A Quantization Method Represented with Ternary and Sparse-Large Weights

## Abstract

Previous ternarizations such as the trained ternary quantization (TTQ), which quantized weights to three values (e.g., $\{-W_n, 0, +W_p\}$), achieved the small model size and efficient inference process. However, the extreme limit on the number of quantization steps causes some degradation in accuracy. To solve this problem, we propose a hybrid weight representation (HWR) method which produces a network consisting of two types of weights, i.e., ternary weights (TW) and sparse-large weights (SLW). The TW is similar to the TTQ's and requires three states to be stored in memory with 2 bits. We utilize the one remaining state to indicate the SLW which is referred to as very rare and greater than TW. In HWR, we represent TW with values while SLW with indices of values by encoding SLW. As a result, the networks can preserve their model size with improving their accuracy compared to ternary weights. To sparsify non-ternary weights in quantization, we also introduce a centralized quantization (CQ) process with a weighted ridge (WR) regularizer. They aim to reduce the entropy of weight distributions by centralizing weights toward ternary values. Our comprehensive experiments show the efficiency of HWR in terms of the trade-off between model size and accuracy.

## 1 Introduction

Deep Neural Networks have made considerable progress in various tasks such as image classification (LeCun et al. 1998, Simonyan & Zisserman 2014, Szegedy et al. 2015), object detection (Ren et al. 2015, Liu et al. 2016), and speech recognition (Graves et al. 2013, Amodei et al. 2016). However, outstanding neural networks usually require deeper and/or wider layers, thus making them hard to deploy on mobile and embedded devices. In response to this problem, many studies set their sights on more efficient networks. Various methods such as pruning (He et al. 2017), light-weights (Howard et al. 2017), and quantization (Courbariaux et al. 2015) have been carried out to reduce the model size and/or computation complexity effectively.

In ternarization, the accuracy degradation is resulted from quantizing values in a limited range with only 2bits. For example, the ternary weights networks (TWN, Li et al. 2016) yields only three quantized values, which prohibits the networks from utilizing high weight values. As known in Han et al. 2015b, large-valued weights tend to have an important role in the prediction. Therefore, the

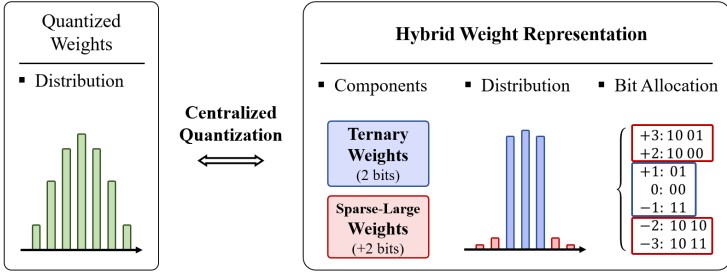

Figure 1: Comparison between conventional quantization and hybrid weight representation (HWR).

absence of large values can cause the accuracy degradation. To solve this problem, our paper proposes a hybrid weight representation (HWR), expressing networks with both ternary weights (TW) and sparse-large weights (SLW). By taking the advantages of both the TW and SLW, the proposed HWR method can preserve their model size compared to ternary weights, as well as avoiding the accuracy degradation in networks.

To be specific, the large values of SLW help networks to improve their accuracy. Furthermore, SLW can be encoded with one remaining state which is not used to store TW in a 2 bits representation. It allows the networks to preserve their model size similarly to ternary weights. The compression rate of the encoding method is affected by the entropy of weight distributions. To train narrower distributions for the efficiency of HWR, we also introduce a centralized quantization (CQ) process and a weighted ridge (WR) regularizer. Figure 1 shows the differences between conventional quantization and HWR. As shown in Figure 1, there is a small number of SLW and the indices of encoded SLW are allocated in storage, unlike TW.

We conduct various experiments, showing that HWR obtains better classification accuracy with the similar model size compared to the trained ternary quantization (TTQ, Zhu et al. 2016), which is a baseline ternarization method. The experiments are carried out on CIFAR-100 (Krizhevsky et al. 2009) and ImageNet (Russakovsky et al. 2015). We use AlexNet (Krizhevsky et al. 2012) and ResNet-18 (He et al. 2016) as baseline networks. Our proposed representation improves the AlexNet performance on CIFAR-100 by 4.15% with only 1.13% increase in the model size.

The contributions of this paper are as follows:

- We propose a hybrid weight representation (HWR), including both values (TW) and indices of values (SLW). SLW allows the networks to improve their accuracy. Besides, the model size can be preserved by encoding SLW with one remaining state of 2 bits for TW.

- We propose a training process, namely centralized quantization (CQ), to improve the efficiency of HWR. In CQ, we can sparsify almost large weights toward ternary weights. The low entropy of the centralized distribution improves the compression rate of encoding.

- We propose a regularizer, namely weighted ridge (WR), which gives more penalty to large weights. WR is utilized to centralize weights for narrower distributions and categorize the weights into TW and SLW.

## 2 RELATED WORK

**Quantization** In low precision training, one major difference from full precision training is that the conventional 32-bits weights ($w$) are discretized by a quantization funtion and represented with the finite number of elements. The discretized weights ($w_q$) are multiplied by input matrices in the feed-forward pass. For example, the binarized neural networks (BNN, Courbariaux & Bengio 2016) utilized a sign function to quantize weights and activations to {-1, +1}. The binarized weights ($w_b$) are defined as:

$$w_b = sign(w), \qquad sign(x) = \begin{cases} +1 & if\ x \geq 0 \\ -1 & otherwise \end{cases}$$

In XNOR-Net (Rastegari et al. 2016), input data is even binarized. Furthermore, the multiplication and addition operations in convolution layers are replaced with XNOR and bit-count operations, respectively. In TWN (Li et al. 2016), the binarized weights are pruned by thresholding as:

$$w_t = \begin{cases} +W_E & if\ w > \Delta \\ 0 & if\ |w| \leq \Delta \\ -W_E & if\ w < -\Delta \end{cases}, \qquad \begin{array}{l} \Delta = 0.7 \cdot E(|w|) \\ W_E = \underset{i \in \{i|\Delta < |w(i)|\}}{E}(|w(i)|) \end{array}$$

In DoReFa-Net (Zhou et al. 2016), they carried out experiments in a wider bit-width and also quantized the gradients. The quantized weights in $k$ bit-width ($w_k$) by a quantization function ($Q_k(\cdot)$) are described as:

$$w_k = 2 \cdot Q_k\left(\frac{tanh(w)}{2max(|tanh(w)|)} + 0.5\right) - 1 , \qquad Q_k(x) = \frac{round(x \cdot (2^k - 1))}{2^k - 1}$$

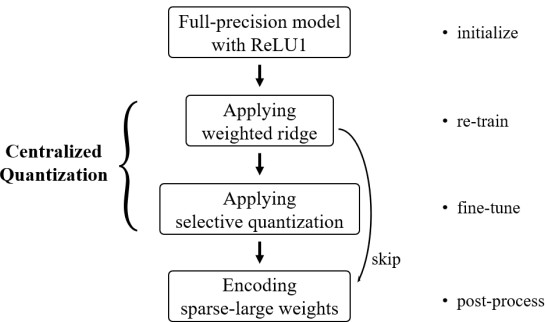

Figure 2: Overall processes of the hybrid weight representation (HWR). The weights are initialized from a full precision model trained with ReLU1 activation. First of the centralized quantization (CQ), the weights are quantized with the weight ridge (WR) as a regularizer. By WR, the quantized weights are also centralized under threshold and categorized into ternary weights (TW) and sparse-large weights (SLW) by threshold. Second, selective quantization (SQ) is applied to fine-tune the weights of the previous step. Finally, SLW is encoded by one usable state of TW as a prefix.

In back-propagation, their is a major concern caused by discretization of full precision weights ($w$). The derivatives of the discretizing functions such as $sign(\cdot)$ and $Q_k(\cdot)$ have zero values at almost input ranges. Therefore, the gradients of $w$, calculated from the discretized weights ($w_b$, $w_t$, and $w_k$), are also zero values and prevent $w$ from optimizing. To solve this vanishing gradient problem, the straight-through estimator (STE) method (Hinton et al. 2012, Bengio et al. 2013) is proposed, where not calculating the gradients of $w$. In other words, the gradients of $w$ ($\frac{\partial L}{\partial w}$) is replaced with the gradients of the discretized weights ($\frac{\partial L}{\partial w_b}$, $\frac{\partial L}{\partial w_t}$, and $\frac{\partial L}{\partial w_k}$) instead of back-propagating zero gradients from discretizing functions.

$$\frac{\partial L}{\partial w} = \frac{\partial L}{\partial w_b} = \frac{\partial L}{\partial w_t} = \frac{\partial L}{\partial w_k}$$

Above methods are linear quantization which have the same intervals between adjoining quantum. By quantizing with the same intervals, the float operations can be replaced to the integer operations.

Their are also non-linear quantization methods that the weights have irregular intervals. For instance, the deep compression (Han et al. 2015a) clustered the weights to quantize and fine-tuned the quantized weights of each clustering group. In TTQ (Zhu et al. 2016), they quantized the weights to ternary values with two trainable scale coefficients for negative and positive weight values, i.e., $\{-W_n, 0, +W_p\}$. In these cases, the model size can be significantly reduced by replacing the weight values with the indices of them. However, the indices are needed to be transformed as weight values. And the irregular intervals make it difficult to utilize integer-based and bit-wise operations.

**Entropy coding** The Huffman coding (Van Leeuwen 1976) aims to compress bit-streams in a lossless manner by reducing bit-length of each element with an optimal prefix tree. The entropy of the weight distribution is one of the important factors that determine the compression rate in Huffman coding. Deep compression (Han et al. 2015a) pruned the weights to make lower entropy so saved more storage when applying the Huffman coding.

**Regularization** The regularization is utilized to manipulate weights in artificial direction by imposing penalty. There are two conventional regularizers such as Ridge (L2) and Lasso (L1) (Han et al. 2015b). The L2 weight decay usually prevents networks from being biased on the training dataset by restricting weights from growing. The penalty of Lasso makes the weights to be close to zero value which are unrelated to predict outputs during training-time. Moreover, the explicit loss of Zhou et al. 2018 makes it possible to quantize weights by controlling the strength of their regularizer. The penalty of regularization can be utilized to change the entropy of the weight distribution.

From previous studies, we focus on that ternary weights, linearly quantized, only require three states when being stored in 2bits, thus the remaining state has a potential to be utilized as a prefix of more extended weights for improving efficiency in quantization. Furthermore, we draw a deduction that the regularization can help us to generate the lower entropy of the weight distribution, maximizing the compression efficiency when encoding the extended weights.

## 3 METHOD

In this section, we explain: i) how centralized quantization (CQ) can centralize weights toward ternary values and categorize the weights into TW and SLW; and ii) how the quantized weights are encoded to be expressed as hybrid weight representation (HWR). The detailed processes are illustrated in Figure 2.

### 3.1 BASIC QUANTIZATION METHOD

The basic quantization (BQ) uses a round function to simply quantize the full precision weights ($w$). The quantization function ($Q_w(\cdot)$) of BQ is fixed for each layer during training-time. Equation 1 shows how $w$ is quantized to $w_q$ by $Q_w(\cdot)$. The $rng$ in Eq 1 is the fixed range of $w$ while low precision training, determined by the maximum absolute value of the pre-trained weights ($M_{w_p} = max(|w_p|)$). Before being entered into $Q_w(\cdot)$, $w$ is clipped to $w_c$ by $rng$ to prevent misquantization of the round function. The $Q_w(\cdot)$ also requires the number of quantization states ($s$) ascertained by the number of bits. To be specific about $Q_w(\cdot)$, the $w_c$ is scaled by a float value to be discretized then it is restored by the reciprocal to the float value after the round function. Using STE, the derivative of $w_c$ ($\frac{\partial L}{\partial w_c}$) is replaced with the derivative of $w_q$ ($\frac{\partial L}{\partial w_q}$). By clipping $w$, we can take a saturation effect on $w$ as in BNN (Courbariaux & Bengio 2016). But if some weights still over $rng$ then $w$ can be restated by clipping again after updating gradients.

$$w_q = Q_w(w_c,\ rng,\ s) = round(w_c \cdot \frac{\frac{s-1}{2}}{rng}) \cdot \frac{rng}{\frac{s-1}{2}}, \qquad w_c = clip(w,\ -rng,\ rng) \tag{1}$$

To quantize activated values, we restrict the range of activated values by using ReLU1 function as activation. The activated values with ReLU1 ($a$) can be quantized to $a_q$ in $k$ bit-width by a round function in Equation 2. As shown in TTQ (Zhu et al. 2016), initializing with the pre-trained model helps the networks to improve their quantization performance. To take this advantage, our training starts with a full precision model in which weights are pre-trained with ReLU1 activation.

$$a = ReLU1(x) = clip(x,\ 0,\ 1) = min(max(0,\ x),\ 1)$$
$$a_q = Q_a(a,\ k) = round(a \cdot (2^k - 1))\ /\ (2^k - 1) \tag{2}$$

### 3.2 WEIGHTED RIDGE

We introduce a new version of L2 weight decay, namely weighted ridge (WR), which aims to achieve two objectives: i) centralizing almost all weights toward below the threshold; and ii) categorizing the weights into TW and SLW by the threshold at the end of re-training.

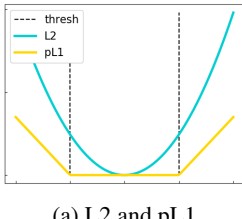

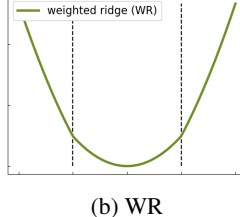

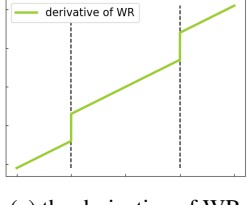

(a) L2 and pL1         (b) WR         (c) the derivative of WR

Figure 3: Specifications of weighted ridge (WR). (a) plots both a normal L2 and a part of L1 (pL1). The WR is an addition of them as in (b). (c) is the derivative of WR.

In this step, we quantize full precision weights ($w$) by the basic quantization (BQ) with WR as a regularizer. This step's quantization function is defined using Equation 1:

$$w_q = Q_w(clip(w,\ -M_{w_p},\ M_{w_p}),\ M_{w_p},\ s_t + s_{sl}) \tag{3}$$

where the $rng$ in Equation 1 is the maximum absolute value of the pre-trained weights ($M_{w_p} = max(|w_p|)$), fixed during re-training. The $s$ in Equation 1 is derived from the number of bits of SLW ($b_{sl}$). If $b_{sl}$ is 2, 3, or 4 bits then the number of quantization steps of SLW ($s_{sl}$) is calculated to 4, 8, or 16 while the number of bits and quantization steps of TW ($b_t$ and $s_t$) are fixed to 2 and 3.

Figure 3 (b) denotes WR which is a mixture of both a normal L2 and a part of L1 (pL1) in (a). The loss of pL1 is only proportionate to the absolute value of the weight which is greater than the threshold ($thresh$). In other words, the large weights receive more penalty compared to the weights under $thresh$ in back-propagation as illustrated in Figure 3 (c). This penalty allows us to sparsify the weights that do not have the necessity to keep their largeness. WR, pL1, and $thresh$ are defined:

$$WR(w, \ \lambda_1, \ \lambda_2) = \lambda_1(L2(w) + \lambda_2 \cdot M_{w_p} \cdot pL1(w)) \tag{4}$$

$$pL1(w) = \begin{cases} |w| - thresh & if \ |w| > thresh \\ 0 & if \ |w| \leq thresh \end{cases}, \quad (thresh = M_{w_p} \cdot \frac{s_t}{s_t + s_{sl} - 1}) \tag{5}$$

There are two hyper-parameters ($\lambda_1$ and $\lambda_2$) which are tools to control the intensity of WR. By adjusting the greater $\lambda_2$, more weights can be sparsified below $thresh$. The $M_{w_p}$ is also a factor for applying different intensity to each layer since some layers have too small $M_{w_p}$ than others.

At the end of applying WR, we classify the re-trained weights into TW and SLW. The $thresh$ is utilized as a criterion for labeling them as Equation 6.

$$mask(w) = \begin{cases} TW & if \ |w| \leq thresh \\ SLW & if \ |w| > thresh \end{cases} \tag{6}$$

An observation is that WR tends to avoid overfitting as effectively as the L2 regularizer. By adjusting WR, we can obtain better accuracy than the L2 weight decay in some experiments as in Table 3.

### 3.3 SELECTIVE QUANTIZATION

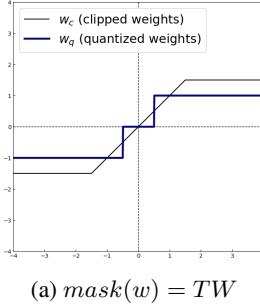
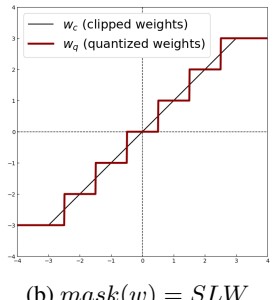

(a) $mask(w) = TW$     (b) $mask(w) = SLW$

Figure 4: In the selective quantization (SQ), two different quantization functions are applied for each category (TW and SLW).

The selective quantization (SQ), the second step of CQ, aims to fine-tune the previous step's weights which are restricted by the additional penalty of the part of L1 (pL1). By removing pL1, the restricted weights can grow again and be more optimized for classification. The key of SQ is that two different quantization functions are applied for each category. As shown in Figure 4 (a), the weights in TW category are always ternarized, thus the percentage of TW is guaranteed during fine-tuning. Otherwise, the weights in SLW category can be quantized to either ternary values or large values as Figure 4 (b). The quantization function of SQ ($Q_s(\cdot)$) can be expressed as Equation 7, modified from Equation 3 (The explanations for $M_{w_p}$, $thresh$, $s_t$, and $s_{sl}$ are in Section 3.2). At the end of fine-tuning, the fine-tuned weights are re-categorized by the same way as Equation 6. This fine-tuning step can be skipped depending on the result of SQ, if it yields only slight difference compared to the weighted ridge (WR).

$$Q_s(w) = \begin{cases} Q_w(clip(w, -thresh, thresh), thresh, s_t + 1) & if \ mask(w) = TW \\ Q_w(clip(w, -M_{w_p}, M_{w_p}), M_{w_p}, s_t + s_{sl}) & if \ mask(w) = SLW \end{cases} \tag{7}$$

Specifically, we use the stochastic gradient descent optimizer (SGD, Bottou 2010) in this fine-tuning step. The momentum optimizer (Sutskever et al. 2013) can misrepresent gradients since the weight values are clipped after updating gradients.

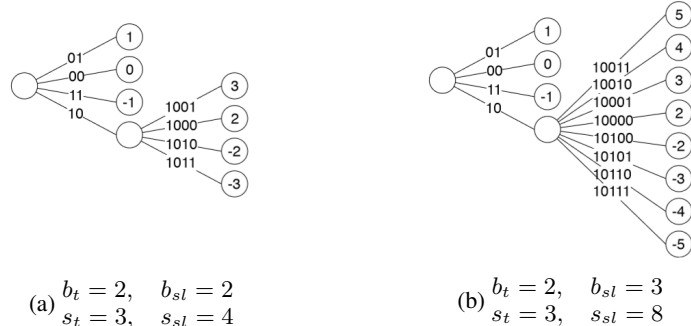

(a) $b_t = 2, \quad b_{sl} = 2$
$s_t = 3, \quad s_{sl} = 4$

(b) $b_t = 2, \quad b_{sl} = 3$
$s_t = 3, \quad s_{sl} = 8$

Figure 5: Encoding trees of hybrid weight representation (HWR). In HWR, we represent TW with values while SLW with indices of values. The number of bits and quantization steps of TW $(b_t, s_t)$ are fixed to 2, 3. In contrast, them of SLW $(b_{sl}, s_{sl})$ can be changed to (2, 4), (3, 8), or (4, 16).

### 3.4 ENCODING SPARSE-LARGE WEIGHTS

The distributions of centralized-quantized weights have low entropy since there are only small number of SLW. It allows the networks to achieve a higher compression rate by encoding SLW. We utilize one remaining state of TW as a prefix to encode SLW as shown in Figure 5. As a result, the hybrid weight representation (HWR) method includes both the values (TW) and the indices (SLW) when the weights are allocated in memory.

There are some advantages of using HWR. The model size can be maintained similar to TTQ (Zhu et al. 2016) since SLW is very rare. The restoration of encoded SLW is not that difficult. The value of SLW can be restored by taking the first sign bit of $b_{sl}$ and adding 2 to the rest of $b_{sl}$. However, there is a concern that HWR causes more complex and extra burden to memory access and inference time due to the SLW. To minimize the negative effect of SLW, we suggest to utilize the sparsity of SLW by gating them or applying a variant of sparse matrix-vector multiplication. Then the TW takes up most of the weights and still has the potential to keep some benefits of ternary weights such as replacing multiplication operations to sign assignment operations (Rastegari et al. 2016) and utilizing gates for skipping zero weights (Deng et al. 2018). More details about our proposed inference methods are in Appendices C.

## 4 EXPERIMENT RESULTS

**Model architecture** We utilize AlexNet (Krizhevsky et al. 2012) and pre-activated ResNet-18 (He et al. 2016) as baseline models, implemented on Tensorflow (Abadi et al. 2016) framework. In AlexNet, we use a variant of it by adding batch normalization (Ioffe & Szegedy 2015) layers and removing dropout layers. In ResNet, we change two things: i) not using skip-connection for the first block of ResNet-18, since the first layer is not quantized and the not quantized weights cannot be entered into the add operations to keep integer-precision stream while inference-time; and ii) mediating of two quantum intervals from convolution layers. In add operations, each input from two convolution layers has different quantum interval due to layer-wise quantization functions. The different intervals make it hard to keep integer-precision stream after add operations. More details of ResNet-18 are explained in Appendices A.

### 4.1 EXPERIMENTS ON CIFAR-100

**Training setup** We set the learning rate of AlexNet to 0.01 with decay 0.1 after every 100 epochs until 300 epochs and the L2 weight decay ($\lambda_1$) to 0.02. We also set the learning rate of ResNet-18 to 0.1 with the same decay of AlexNet and the $\lambda_1$ to 0.0002. For the dataset, we select CIFAR-100 Krizhevsky et al. 2009) and set the minibatch size to 128 to estimate the performance of networks. We also use Nesterov Accelerated Gradients (Sutskever et al. 2013) as an optimizer. To prevent the fluctuation of validation accuracy, we take a simple moving averages (sma) in 5 epochs during training-time and average the sma over four repetitions. All experiments are our implementation.

Table 1: The results of both the top-1 accuracy and the average bit length in multiple bit-width. The full experiments of ResNet-18 are listed in Appendices E

| Model Name | Bit-W $(b_t/b_{sl})$ | Bit-A | $\lambda_2$ | Percentage of SLW | Average Bit Length | Top-1 Accuracy | Remarks |
|---|---|---|---|---|---|---|---|
| AlexNet | 32 | 32 | - | - | - | 75.06% | FP |
| AlexNet | 32 | 32 | - | - | - | 74.78% | FP(ReLU1) |
| AlexNet | 2 / - | 4 | - | 0% | 2 | 70.57% | TTQ* |
| AlexNet | 2 / 3 | 4 | - | 12.05% | 2.362 | 73.54% | BQ |
| AlexNet | 2 / 3 | 4 | 1.2 | 1.39% | 2.042 | 73.39% | BQ+WR |
| AlexNet | 2 / 3 | 4 | - | 0.83% | 2.025 | **74.72%** | SQ |
| ResNet-18 | 32 | 32 | - | - | - | 76.07% | FP |
| ResNet-18 | 32 | 32 | - | - | - | 74.54% | FP(ReLU1) |
| ResNet-18 | 2 / - | 4 | - | 0% | 2 | 72.79% | TTQ* |
| ResNet-18 | 2 / 2 | 4 | - | 0.59% | 2.0118 | 73.92% | BQ |
| ResNet-18 | 2 / 2 | 4 | 1.0 | 0.014% | 2.0003 | 73.92% | BQ+WR |
| ResNet-18 | 2 / 2 | 4 | - | 0.0015% | 2.00003 | **74.22%** | SQ |
| ResNet-18 | 2 / 3 | 4 | - | 7.34% | 2.2203 | 74.07% | BQ |
| ResNet-18 | 2 / 3 | 4 | 1.0 | 0.434% | 2.013 | 74.05% | BQ+WR |
| ResNet-18 | 2 / 3 | 4 | - | 0.321% | 2.0096 | **74.67%** | SQ |
| ResNet-18 | 2 / 4 | 4 | - | 28.80% | 3.1521 | 74.54% | BQ |
| ResNet-18 | 2 / 4 | 4 | 1.0 | 4.995% | 2.2 | 74.32% | BQ+WR |
| ResNet-18 | 2 / 4 | 4 | - | 4.145% | 2.166 | **74.84%** | SQ |

**Implementation** Our experiments show the comparison between our proposed methods (BQ, BQ with WR, and SQ in Section 3) and TTQ (Zhu et al. 2016) as a baseline model. In weight quantization, the number of bits of TW ($b_t$) is fixed to 2 while the number of bits of SLW ($b_{sl}$) can be changed to 2, 3, or 4. In activation quantization, we select 4 bits to quantize the activations. For the original TTQ paper, they did not quantize activated values. In our implementation of TTQ (TTQ*), we quantize activations with the same quantization function in Equation 2 to make similar condition. Also, our ResNet-18 architecture for TTQ* can be different from the origian TTQ as written above. We quantize all the weights and activations, even the last activation before the global average pooling, except the weights of first and last layers and the outputs of soft-max layer.

In Table 1, the effect of the weighted ridge (WR) regularizer can be found out from the comparison of the basic quantization (BQ) and BQ with WR. The results of BQ+WR show more centralized distribution and less accuracy than only BQ. Furthermore, we observe that the results of the selective quantization (SQ) have more centralized distribution than BQ+WR and more improved accuracy than BQ, removing the part of L1 in WR and fine-tuning the restricted weights by WR. After the encoding step, the difference of the average bit length between SQ and TTQ* becomes similar. When $b_{sl}$ is 3 for AlexNet, SQ reaches top-1 accuracy of 74.72%, which is 4.15% higher than TTQ*, with only 1.25% increase in the model size. In ResNet-18, when $b_{sl}$ is 3, the top-1 accuracy of SQ is 1.88% higher than TTQ* with only 0.5% loss in the model size.

## 4.2 EXPERIMENTS ON IMAGENET

**Training setup** We performed the experiments of ResNet-18 on ImageNet (Russakovsky et al. 2015) dataset. We set the learning rate to 0.1 with decay 0.1 after 25, 50, 75, and 80 epochs until 85 epochs and the minibatch size to 256. We use the momentum optimizer (Sutskever et al. 2013). We take a simple moving average of validation error in 5 epochs during training-time.

In Table 2, the results of BQ, BQ+WR, and SQ on ImageNet show a similar tendency of the results on CIFAR-100. The SQ ($b_{sl} = 4$) reaches top-1 accuracy of 59%, which is 6.41% higher than TTQ*,

Table 2: The results of both the top-1 accuracy and the average bit length on ImageNet dataset.

| Model Name | Bit-W $(b_t/b_{sl})$ | Bit-A | $\lambda_2$ | Percentage of SLW | Average Bit Length | Top-1 Accuracy | Remarks |
|---|---|---|---|---|---|---|---|
| ResNet-18 | 32 | 32 | - | - | - | 70.41% | FP |
| ResNet-18 | 32 | 32 | - | - | - | 62.51% | FP(ReLU1) |
| ResNet-18 | 2 | 32 | - | 0% | 2 | 66.6% | TTQ |
| ResNet-18 | 2 | 4 | - | 0% | 2 | 52.59% | TTQ* |
| ResNet-18 | 2 / 3 | 4 | - | 1.11% | 2.033 | 55.51% | BQ |
| ResNet-18 | 2 / 3 | 4 | 0.25 | 0.84% | 2.025 | 55.20% | BQ+WR |
| ResNet-18 | 2 / 3 | 4 | - | 0.54% | 2.016 | **57.67%** | SQ |
| ResNet-18 | 2 / 4 | 4 | - | 4.012% | 2.161 | 58.43% | BQ |
| ResNet-18 | 2 / 4 | 4 | 1 | 0.293% | 2.012 | 56.70% | BQ+WR |
| ResNet-18 | 2 / 4 | 4 | - | 0.269% | 2.011 | **59.00%** | SQ |

with only 0.55% loss in the average bit length. One notable point is that using ReLU1 activation causes large accuracy degradation, even the accuracy of TTQ (Zhu et al. 2016) is higher than the full precision model with ReLU1.

### 4.3 Ablation study

Table 3: The results of the weighted ridge (WR) regularizer on full precision models.

| Model Name | Quantizaton Plan $(b_t/b_{sl})$ | $\lambda_2$ | Top-1 Accuracy | Remarks |
|---|---|---|---|---|
| AlexNet | - | - | 75.06% | FP |
| AlexNet | 2 / 3 | 1.2 | **75.32%** | FP+WR |
| ResNet-18 | - | - | 76.07% | FP |
| ResNet-18 | 2 / 2 | 1 | **76.49%** | FP+WR |
| ResNet-18 | 2 / 3 | 1 | **76.28%** | FP+WR |
| ResNet-18 | 2 / 4 | 1 | **76.88%** | FP+WR |

In this study, we try to observe the effect of the weighted ridge (WR) regularizer on full precision models as shown in Table 3. When $\lambda_2$ is set to 1, the penalty of the part of L1 (pL1) of WR is calculated to the maximum absolute value of the pre-trained model ($M_{w_p}$) by the derivative of Equation 4, which is sufficiently large because the penalty $M_{w_p}$ is same with the maximum penalty of L2 ($M_{w_p}$). The results of AlexNet and ResNet-18 with WR obtain 0.26% and 0.81% higher accuracy than the base-line models. The pL1 prevents the networks from overfitting as effectively as L2 regularizer. We conjecture that the biased large weights toward training dataset have more negatively effect on generalization than small weights. Therefore, the restriction of large weights enables us to make more generalized models. More ablation studies are in Appendices B.

## 5 Discussion

### 5.1 Trade-off between accuracy and model size

We perform more extended experiments in various $\lambda_2$ of WR to monitor the correlation between the percentage of SLW and the accuracy of ResNet-18 on CIFAR-100. The complete results are listed in Appendices E. Figure 6 shows that the intensity of pL1 ($\lambda_2$) in Equation 4 has less relation with the classification accuracy. In spite of the greater $\lambda_2$, some results show even better accuracy than

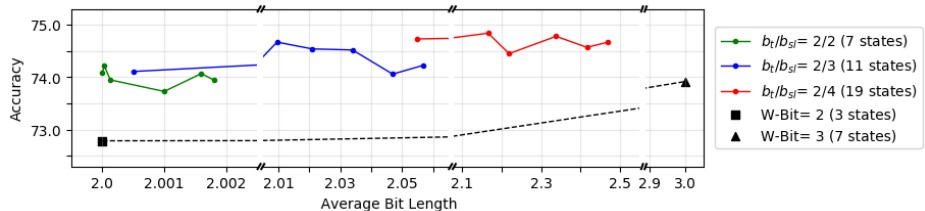

Figure 6: The trade-off between accuracy and model size in various $\lambda_2(2, 1, \frac{1}{2}, \frac{1}{4}, \frac{1}{8}, \frac{1}{16})$.

lower $\lambda_2$. Therefore, we can perceive that many large weights in networks can be centralized with comparable accuracy, accompanied of generalization.

## 5.2 THE WEIGHT DISTRIBUTIONS

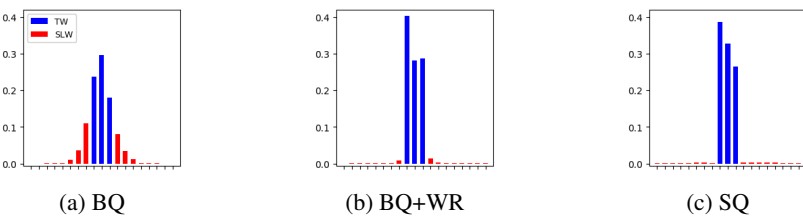

Figure 7: The weight distributions of each method.

As illustrated in Figure 7, the blue and red bars refer to the percentage of TW and SLW respectively. In the SQ step, the forcibly restricted weights by WR can grow during fine-tuning, resulting in less percentage of SLW than BQ+WR as in Table 1. From that, we assume that when a weight starting to grow in fine-tuning begins to contribute more importantly for prediction than some other weights which have made a similar contribution with the growing weight, the other weights become smaller in contrast. The most central weights of SLW in BQ+WR are wide-spread after SQ step. This shows that the penalty of pL1 cannot be proper to make more naturally centralized distributions. To make more flatten distributions after centralization, we suggest using another version of WR which utilize a part of exponential lasso (pEL1, Breheny 2015) instead of pL1 as written in Appendices D.

## 5.3 ANALYSIS OF QUANTIZATION METHOD.

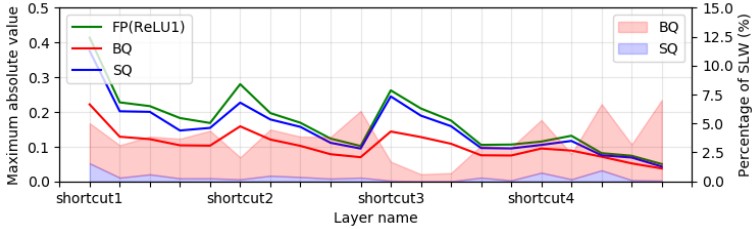

Figure 8: Layer-wise comparison of the percentage of SLW and the maximum absolute value.

Figure 8 displays the layer-wise maximum absolute value and percentage of SLW. If one layer has many SLW after BQ then the layer tends to have a relatively higher number of SLW after SQ. The maximum absolute values (max-abs) of SQ tend to be larger than them of BQ since the restriction of the number of SLW causes more strong gradients for large weights. The max-abs of BQ are remarkably lower than the full precision model, used to fix the range of quantization functions at initialization. That shows a problem that the basic quantization (BQ) method is not an effective way because some layers do not use the whole quantization states. For this reason, we need to apply more effective quantization methods instead of BQ. The following Table 4 lists other methods.

Table 4: Classification accuracy of the state-of-the-art quantization methods trained on ImageNet with Resnet-18 model. Bit-W and Bit-A refer to the weights and activations bit-width respectively.

| Model | Bit-W | Bit-A | Top-1 Accuracy |
|---|---|---|---|
| HWR (ours, $b_t/b_{sl} = 2/3$) | 2/3 | 4 | 57.67% |
| HWR (ours, $b_t/b_{sl} = 2/4$) | 2/4 | 4 | 59.00% |
| ABC-Net (Lin et al. 2017) | 3 | 3 | 61.0% |
| PACT (Choi et al. 2018) | 4 | 4 | 69.2% |
| QIL (Jung et al. 2019) | 4 | 4 | 70.1% |

**The state-of-the-art quantizations** Table 4 presents that our method HWR has less accuracy than other state-of-the-art quantization methods. There are two main reasons: i) more complex quantization methods. For example, PACT (Choi et al. 2018) use parameterized clipping activation and QIR (Jung et al. 2019) use a transformer before discretizing weights; ii) differences of detail of ResNet-18 architecture. As explained in Section 4, we try to keep the integer-precision stream during inference-time. Especially, the skip-connections and add operations of ResNet can disturb the stream to maintain integer-precision since two outputs from different convolution layers, having different quantization intervals, should be added.

To deal with that BQ has low performance, the state-of-the-art quantization methods can be harmonized with the centralized quantization (CQ) process. The full precision model for initialization can be replaced with the pre-quantized model trained by those methods. The quantization function of BQ can be replaced with the quantization function of the pre-quantized model.

## 6 CONCLUSION

We propose the hybrid weight representation (HWR), consisting of two types of weights such as the ternary weights (TW) and sparse-large weights(SLW). In HWR, we represent TW with values while SLW with indices of values. To represent SLW as indices, we encode SLW by utilizing one usable state as a prex, which is not used to store TW in 2 bits. To maximize the effect of encoding, we introduce the centralized quantization (CQ) process: i) applying the basic quantization (BQ) with the weighted ridge (WR) as a regularizer to centralize and categorize the weights; and ii) applying the selective quantization (SQ) to fine-tune the weights. After encoding process, the model size of HWR is similar with the TTQ even non-ternary weights exist. Additionally, we propose a method to mediate two different quantum intervals of add operations in ResNet to keep the integer-precision stream while inference-time. As a result, SLW helps the networks to improve their accuracy while preserving their model size by encoding SLW compared to the 2-bit representation. We can achieve more efficient networks in terms of the trade-off between accuracy and model size.

## 7 FUTURE WORK

We consider three ways to extend our research: i) using more accurate quantization method instead of BQ. The pre-quantized weights and quantization functions from state-of-the-art methods can be utilized for HWR; ii) using more bits for central weights. In mid-tread and symmetrical quantization, there is still one remaining state similarly with the ternary weights. Therefore, more experiments can be carried out to find the optimal number of bits for center and edge weights; and iii) verifying the performance of suggested inference methods. As written in Appendices C, we suggest efficient inference methods utilizing the sparsity of SLW to minimize the extra burden caused by SLW.

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

# A DETAILS OF RESNET

## A.1 THE MEDIATION OF QUANTUM INTERVALS

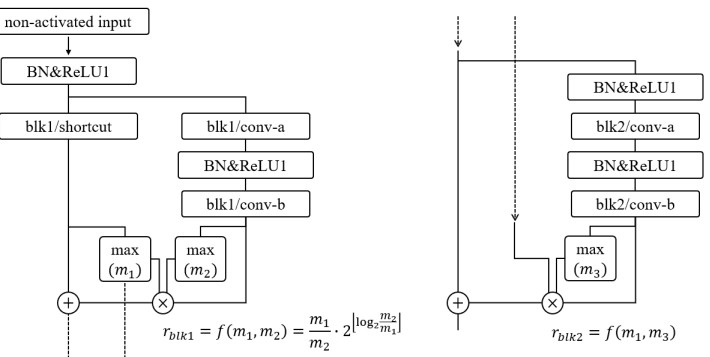

Figure 9: Before add operations in ResNet, the outputs of last convolution layers from each block are scaled by a scale factor $r$, derived from the maximum absolute values of convolution layers.

In our quantization method, the weights and activations are quantized in the same intervals ($w_q = i_w \cdot I_w$, $a_q = i_a \cdot I_a$). The $I_w$ and $I_a$ are fixed float intervals while the $i_w$ and $i_a$ are integer variables. Therefore, convolution and fully-connected layers can be inferred by integer operations as Equation 8. Other operations such as batch normalization (BN, Ioffe & Szegedy 2015), ReLU1 activation, and the $Q_a(\cdot)$ also can be integrated and compressed by integer comparators (Lahoud et al. 2019, Lin et al. 2017) since the coefficients of BN and boundaries of ReLU1 and $Q_a(\cdot)$ are fixed so can be prearranged.

$$o_q = layer(a_q, \ w_q) = \sum \sum a_q w_q = I_a I_w \cdot \sum \sum i_a i_w \qquad (8)$$

However, ResNet (He et al. 2016) has identity mappings and add operations that prevent the data streams from keeping the same quantum intervals due to the layer-wise intervals. To deal with this problem, we scaled one input of add operations by a scale factor $r$ as defined in Equation 9. The $r$ is derived from the maximum absolute values of the shortcut ($m_{short}$) and last convolution ($m_{conv_b}$) layers from each block as in Figure 9. Since we use a log and floor function, $r$ is always in the range (0.5, 1] and it helps to keep the effect of identity mapping of ResNet. If $r$ is zero then nothing is added in the add operation or if $r$ is larger than 1 then the shortcut can be perturbed by a large residual.

$$r = f(m_{short}, m_{conv_b}) = \frac{m_{short}}{m_{conv_b}} \cdot 2^{\left\lfloor \log_2 \frac{m_{conv_b}}{m_{short}} \right\rfloor}, \qquad (0.5 < r \le 1) \qquad (9)$$

By adjusting $r$, two different intervals from $j$th ($shortcut$) and $k$th ($conv_b$) convolution layers can be mediated and it will allow us to add two inputs with only shift operations (multiplication of multiple of 2) as Equation 10.

$$add(o_q^j, o_q^k) = m_{short} \cdot (2^{-\left\lfloor \log_2 \frac{I_a^k I_w^k}{I_a^j I_w^j} \right\rfloor} \cdot \sum \sum i_a^j i_w^j + \sum \sum i_a^k i_w^k) \qquad (10)$$

## A.2 THE FIRST RESIDUAL BLOCK OF RESNET

In our quantization, the first convolution layer is not quantized. Therefore, to prevent ResNet from having mixed precision output after add operations, our first residual block do not have a skip-connection even though the input shape of the block is same with the output shape of the first layer.

# B  ABLATION STUDY

## B.1  FIXING THE RANGE OF THE QUANTIZATION FUNCTION

Table 5: Comparison of non-fixing the range of the quantization function (TTQ*) and fixing them (TTQ**).

| Model Name | Bit-W | Bit-A | Top-1 Accuracy | Remarks |
|---|---|---|---|---|
| AlexNet | 32 | 32 | 75.06% | FP |
| AlexNet | 2 | 4 | 70.57% | TTQ* |
| AlexNet | 2 | 4 | 72.04% | TTQ** |
| ResNet-18 | 32 | 32 | 76.07% | FP |
| ResNet-18 | 2 | 4 | 72.79% | TTQ* |
| ResNet-18 | 2 | 4 | 73.55% | TTQ** |

To observe the effect of the fixed range of the basic quantization (BQ) method, we perform a particular experiment. The results of our implementation of TTQ (TTQ*) is performed with the quantization function of which the range is not fixed. On the other hand, a version of implementation (TTQ**) have two different conditions compared to TTQ*: i) fixing the range of the quantization function at initialization; and ii) using one coefficient as $\{-W_s, 0, +W_s\}$. When applying them, the accuracy of AlexNet and ResNet-18 are rather improved by 1.47% and 0.76%. The results of TTQ** shows that the fixed range also can be utilized for quantization. Additionally, TTQ set a constant $t$ to 0.05 for quantizing weights under $t \cdot max(|w|)$ as zero values. In our method, however, if $b_{sl}$ is 2, 3, or 4 bits then the $t$ is derived as $\frac{1}{6}$, $\frac{1}{10}$, or $\frac{1}{18}$. All our $t$ are higher than 0.05 of TTQ and it can cause worse results since L2 regularizer gives more penalty to the larger weights.

## B.2  MEDIATION OF QUANTUM INTERVALS

Table 6: The results of applying the scaling factor $r$, derived from the maximum absolute values of two inputs of add operations, to mediate two different intervals on ResNet-18 with full precision.

| Model Name | Scaling | Activation function | $\lambda_1$ | Top-1 Accuracy |
|---|---|---|---|---|
| ResNet-18 | - | ReLU | 0.0002 | 76.07% |
| ResNet-18 | $\frac{m_{short}}{m_{conv_b}}$ | ReLU | 0.0002 | 75.80% |
| ResNet-18 | $r$ | ReLU | 0.0002 | **76.15%** |
| ResNet-18 | - | ReLU1 | 0.0002 | 74.54% |
| ResNet-18 | $\frac{m_{short}}{m_{conv_b}}$ | ReLU1 | 0.0002 | 74.16% |
| ResNet-18 | $r$ | ReLU1 | 0.0002 | **74.54%** |

In Table 6, the experiments are conducted to observe the effect of the coefficient $r$, introduced in Appendices A.1, on the accuracy of ResNet-18. We compare three coefficients such as zero, $r$ in Equation 9, and $m_{short}/m_{conv_b}$. We observe that $r$ performs as similar as non-scaled models. As mentioned in Appendices A.1, the $m_{short}/m_{conv_b}$, greater than 1, can perturb the shortcut connections. We apply $r$ scaling for all our quantization experiments on ResNet-18.

# C  EFFICIENT INFERENCE METHODS

In this section, we suggest two methods for efficient inference of HWR. Those methods utilize the sparsity of SLW to minimize the extra burden caused by SLW and the benefit of ternary weights that

the multiplication of ternary weights can be replaced with the sign assignment operation (Rastegari et al. 2016).

## C.1 GATED MULTIPLICATION

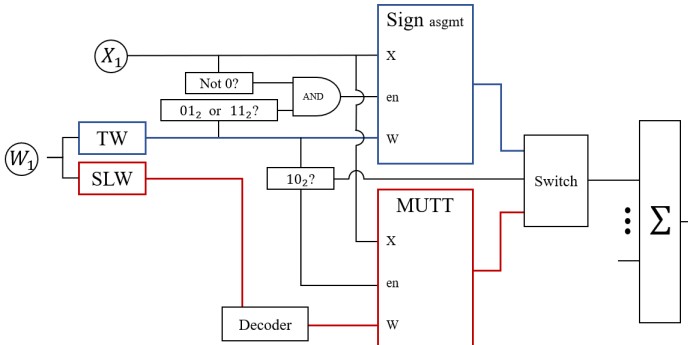

Figure 10: Computing architecture for HWR in inference time. According to the TW of each weight $(W_1)$, the operation for input $(X_1)$ is determined. There are two operations such as sign assignment (sign asgmt) and integer multiplication (MUTT) for each type, TW and SLW.

In GXNOR-Net (Deng et al. 2018), they proposed gated XNOR architecture to take an advantage of binary weights when using ternary weights. If the input or weight are zero then the multiplication can be skipped while if they are not zero then the XNOR operation can be applied similarly with the XNOR-Net (Rastegari et al. 2016). In HWR case, we use enable pins as gates to divide two cases of TW and SLW. As shown in Figure 10, if the TW is $01_2$ or $11_2$ and input is not zero then only sign assignment operation (sign asgmt) is done while if the TW is $10_2$ then only integer multiplication (MUTT) is done. The sign-assigned or multiplied values are added at the end to be an output of one neuron.

## C.2 SEPARATED MULTIPLICATION

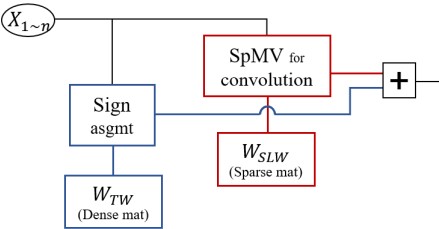

Figure 11: Computing architecture for HWR in inference time. There are two weight matrices such as a dense matrix $W_{TW}$ and a sparse matrix $W_{SLW}$. The original weight matrix can be restored by sum of $W_{TW}$ and $W_{SLW}$. In inference time, sign assignment operation is carried out with $W_{TW}$ while sparse convolution is carried out with $W_{SLW}$. The output of a layer is sum of outputs from two operations.

In conventional layer, there is only one weight matrix. On the other hand, we can divide the one quantized matrix $(W_q)$ to two matrices such as a dense matrix $W_{TW}$ and a sparse matrix $W_{SLW}$ as in Equation 11.

$$W_q = W_{TW} + W_{SLW} \tag{11}$$

The $W_{TW}$ and $W_{SLW}$ can be described as:

$$W_{TW} = \begin{cases} W_q & if\ mask(w) = TW \\ 0 & otherwise \end{cases}, \quad W_{SLW} = \begin{cases} 0 & otherwise \\ W_q & if\ mask(w) = SLW \end{cases} \tag{12}$$

In inference time, the multiplication of quantized activations ($A_q$) and quantized weights ($W_q$) also can be divided as:

$$A_q \cdot W_q = A_q \cdot W_{TW} + A_q \cdot W_{SLW} \qquad (13)$$

For multiplication of $A_q$ and $W_{TW}$, the only sign assignment is required. For multiplication of $A_q$ and $W_{SLW}$, we can utilize sparse convolution operation (Park et al. 2016) to reduce computational complexity. After applying each operation, two calculated results are added to be an output of a layer.

## D    A PART OF EXPONENTIAL LASSO

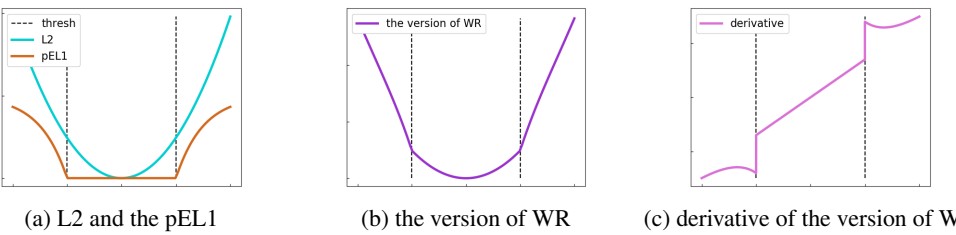

(a) L2 and the pEL1        (b) the version of WR        (c) derivative of the version of WR

Figure 12: Another version of WR, using the part of exponential lasso (pEL1) instead of pL1. (a) denotes both L2 and pEL1. The version of WR is addition of them as in (b). (c) is the derivative of the version of WR.

In Figure 7 (b), the most central weights in SLW has a particularly high probability than other weights in SLW. This is because the penalty of pL1 prevents the weights from being large. If the penalty of pL1 for the weights, large enough for $thresh$, is reduced then the particularly centralized weights would grow well. For that, we suggest a version of WR which utilize a part of exponential L1 (pEL1, Breheny 2015) instead of pL1. The version of WR (WR*) and pEL1 can be defined as Equation 14, 15. The $\tau$ is an additional tuning parameter.

$$WR^*(w, \; \lambda_1, \; \lambda_2) = \lambda_1 (L2(w) + \lambda_2 \cdot M_{w_p} \cdot pEL1(w)) \qquad (14)$$

$$pEL1(w) = \begin{cases} \frac{\lambda_2}{\tau} \left\{ 1 - exp\left( -\frac{\tau}{\lambda_2} \cdot (|w| - thresh) \right) \right\} & if \; |w| > thresh \\ 0 & if \; |w| \leq thresh \end{cases} \qquad (15)$$

## E    FULL EXPERIMENTS OF RESNET-18

Table 7 shows the full version of our experiment results on CIFAR-100 dataset with ResNet-18 model in various $\lambda_2$.

Table 7: The experimental results of HWR and CQ in multiple bit-width and various $\lambda_2$.

| Model Name | Bit-W $(b_t/b_{sl})$ | Bit-A | $\lambda_2$ | Percentage of SLW | Average Bit Length | Top-1 Accuracy | Remarks |
|---|---|---|---|---|---|---|---|
| ResNet-18 | 32 | 32 | - | - | - | 76.07% | FP |
| ResNet-18 | 32 | 32 | - | - | - | 74.54% | FP(ReLU1) |
| ResNet-18 | 2 / - | 4 | - | 0% | 2 | 72.79% | TTQ* |
| ResNet-18 | 2 / 2 | 4 | - | 0.59% | 2.0118 | 73.92% | BQ |
| ResNet-18 | 2 / 2 | 4 | 0.0625 | 0.373% | 2.0075 | 73.57% | BQ+WR(1) |
| ResNet-18 | 2 / 2 | 4 | - | 0.092% | 2.0018 | **73.95%** | SQ(1) |
| ResNet-18 | 2 / 2 | 4 | 0.125 | 0.198% | 2.004 | 73.93% | BQ+WR(2) |
| ResNet-18 | 2 / 2 | 4 | - | 0.079% | 2.0016 | **74.07%** | SQ(2) |
| ResNet-18 | 2 / 2 | 4 | 0.25 | 0.095% | 2.002 | 73.73% | BQ+WR(3) |
| ResNet-18 | 2 / 2 | 4 | - | 0.049% | 2.001 | **73.73%** | SQ(3) |
| ResNet-18 | 2 / 2 | 4 | 0.5 | 0.023% | 2.0004 | 74.11% | BQ+WR(4) |
| ResNet-18 | 2 / 2 | 4 | - | 0.0068% | 2.00013 | **73.95%** | SQ(4) |
| ResNet-18 | 2 / 2 | 4 | 1.0 | 0.014% | 2.0003 | 73.92% | BQ+WR(5) |
| ResNet-18 | 2 / 2 | 4 | - | 0.0015% | 2.00003 | **74.22%** | SQ(5) |
| ResNet-18 | 2 / 2 | 4 | 2.0 | 0.0002% | 2.000004 | 73.93% | BQ+WR(6) |
| ResNet-18 | 2 / 2 | 4 | - | 0.0004% | 2.0+8e-7 | **74.08%** | SQ(6) |
| ResNet-18 | 2 / 3 | 4 | - | 7.34% | 2.2203 | 74.07% | BQ |
| ResNet-18 | 2 / 3 | 4 | 0.0625 | 4.955% | 2.149 | 74.19% | BQ+WR(1) |
| ResNet-18 | 2 / 3 | 4 | - | 1.904% | 2.057 | **74.23%** | SQ(1) |
| ResNet-18 | 2 / 3 | 4 | 0.125 | 3.393% | 2.102 | 74.05% | BQ+WR(2) |
| ResNet-18 | 2 / 3 | 4 | - | 1.563% | 2.047 | **74.06%** | SQ(2) |
| ResNet-18 | 2 / 3 | 4 | 0.25 | 2.11% | 2.063 | 74.28% | BQ+WR(3) |
| ResNet-18 | 2 / 3 | 4 | - | 1.147% | 2.034 | **74.52%** | SQ (3) |
| ResNet-18 | 2 / 3 | 4 | 0.5 | 0.967% | 2.029 | 74.1% | BQ+WR(4) |
| ResNet-18 | 2 / 3 | 4 | - | 0.7% | 2.021 | **74.54%** | SQ(4) |
| ResNet-18 | 2 / 3 | 4 | 1.0 | 0.434% | 2.013 | 74.05% | BQ+WR(5) |
| ResNet-18 | 2 / 3 | 4 | - | 0.321% | 2.0096 | **74.67%** | SQ(5) |
| ResNet-18 | 2 / 3 | 4 | 2.0 | 0.018% | 2.0006 | 74.3% | BQ+WR(6) |
| ResNet-18 | 2 / 3 | 4 | - | 0.016% | 2.0005 | **74.11%** | SQ(6) |
| ResNet-18 | 2 / 4 | 4 | - | 28.80% | 3.1521 | 74.54% | BQ |
| ResNet-18 | 2 / 4 | 4 | 0.0625 | 20.67% | 2.827 | 74.56% | BQ+WR(1) |
| ResNet-18 | 2 / 4 | 4 | - | 11.749% | 2.47 | **74.67%** | SQ(1) |
| ResNet-18 | 2 / 4 | 4 | 0.125 | 16.608% | 2.664 | 74.28% | BQ+WR(2) |
| ResNet-18 | 2 / 4 | 4 | - | 10.395% | 2.416 | **74.57%** | SQ(2) |
| ResNet-18 | 2 / 4 | 4 | 0.25 | 12.38% | 2.495 | 74.53% | BQ+WR(3) |
| ResNet-18 | 2 / 4 | 4 | - | 8.462% | 2.338 | **74.78%** | SQ(3) |
| ResNet-18 | 2 / 4 | 4 | 0.5 | 8.108% | 2.324 | 74.26% | BQ+WR(4) |
| ResNet-18 | 2 / 4 | 4 | - | 5.485% | 2.219 | **74.45%** | SQ(4) |
| ResNet-18 | 2 / 4 | 4 | 1.0 | 4.995% | 2.2 | 74.32% | BQ+WR(5) |
| ResNet-18 | 2 / 4 | 4 | - | 4.145% | 2.166 | **74.84%** | SQ(5) |
| ResNet-18 | 2 / 4 | 4 | 2.0 | 1.611% | 2.064 | 74.06% | BQ+WR(6) |
| ResNet-18 | 2 / 4 | 4 | - | 1.381% | 2.055 | **74.73%** | SQ(6) |

