# OpenReview forum: "Hybrid Weight Representation: A Quantization Method Represented with Ternary and Sparse-Large Weights"
_ICLR.cc/2020/Conference — Reject_

### Official Review · AnonReviewer3 · 2019-10-23
**Official Blind Review #3**

**Rating:** 3

**Review:**

Summary:

The paper proposes a hybrid weights representation method where the weights of the neural network is split into two portions: a major portion of ternary weights and a minor portion of weights that are represented with different number of bits. The two portions of weights are differentiated by using the previous unused state of a typical ternary neural network since only three states are used out of the four states given 2-bit representation. The experiments are solid based on the selected baseline model on CIFAR-100 and Imagenet dataset.

Pros:

•	The idea of using the previous unused state in ternary neural network is interesting
•	Overall, the paper is well written. The proposed method is presented clearly with proper graph illustration.
Cons:
•	The idea of using mixed bit width for neural network quantization is not new. However, the experiments in the paper only compare with basic quantization method which makes the comparison not fair enough. For example, in ABC-net[1], a few full precision coefficients are used to binarize the network. With 3 bit for both weights and activations, it achieves 61% top 1 classification on ImageNet dataset with ResNet-18 as backbone model. This is around 3% higher than the paper’s proposed method with 2/4 bits for weights and 4 bits for activations.
•	In the paper, it claims that the proposed weight ridge method “can obtain better accuracy than L2 weights decay”. However, there are no experiments or any theoretical supports for it.
•	After utilizing the forth state of a ternary neural network, it implies that all four states provided by 2 bit representation are used. Hence, the comparison with a quantized neural network of 2 bits should be given in the experiments also.

[1] Lin, Xiaofan, Cong Zhao, and Wei Pan. "Towards accurate binary convolutional neural network." Advances in Neural Information Processing Systems. 2017.


**Experience Assessment:**

I have published in this field for several years.

**Review Assessment: Checking Correctness Of Derivations And Theory:**

I assessed the sensibility of the derivations and theory.

**Review Assessment: Checking Correctness Of Experiments:**

I assessed the sensibility of the experiments.

**Review Assessment: Thoroughness In Paper Reading:**

I read the paper at least twice and used my best judgement in assessing the paper.

---

> ### Author Response · Authors · 2019-11-06
> **First Replies**
>
> Thank you for giving us a chance to improve our paper. We also agree with your concerns and hope our replies would work for you.
>
> 1) The experiments in the paper only compare with the basic quantization method which makes the comparison not fair enough.
> >> This is a really great question. Actually, we concentrated on the effect of centralized quantization using sparse-large weights. As you mentioned, there are many state-of-the-art quantization methods. To minimize some unexpected or different effects by using state-of-the-art quantization methods such as PACT[1] or QIL[2] when evaluating the centralized quantization, we selected a simple quantization method which is the Basic Quantization (Sec 3.1). Our future work is that applying the centralized quantization to other state-of-the-art quantization methods. As you mentioned, the quantization methods of PACT[1] and QIL[2] is better than Basic Quantization (Sec 3.1). In a reflection of your advice, we will add other state-of-the-art results. Can you recommend a proper section to add the contents of state-of-the-art quantization methods? i) related work, ii) Sec 3.1 (Basic quantization), ii) experimental results (like ABC-net[3]), or iv) multiple choice.
>
> 2) There are no experiments or any theoretical supports for the proposed weight ridge method.
> >> We think that our presentation was unclear. Table 1 and Sec 4.1 show the ablation study investigating the effect of the weighted ridge in full-precision models.
> Like your opinion, we will separate the main experimental results (table 3, 4) and ablation studies (table 1, 2).
>
> 3) The comparison with a quantized neural network of 2 bits should be given in the experiments also.
> >> It seems a considerable point. In my understanding, you suggest that experiments of a quantized neural network of 2 bits should be given. Actually, we do not consider the quantized neural network of 2 bits because we cannot keep the benefit of ternary weights. Ternary weights do not require multiply operations in inference-time, while 2-bits weights require multiply operations. And there is a potential to keep this benefit when using our hybrid representation method by applying the variant of sparse convolution operations[4]. We will add more details about inference process to appendices.
>
> We'll update your addressed points. If you want, we can remind you after revising by adding a comment.
>
> If you have more ambiguous terms or any questions, please claim those for revising our paper.
>
> Best Regards.
>
> [1] Choi, Jungwook, et al. "Pact: Parameterized clipping activation for quantized neural networks." arXiv preprint arXiv:1805.06085 (2018).
> [2] Jung, Sangil, et al. "Learning to quantize deep networks by optimizing quantization intervals with task loss." Proceedings of the IEEE Conference on Computer Vision and Pattern Recognition. 2019.
> [3] Lin, Xiaofan, Cong Zhao, and Wei Pan. "Towards accurate binary convolutional neural network." Advances in Neural Information Processing Systems. 2017.
> [4] Park, Jongsoo, et al. "Faster cnns with direct sparse convolutions and guided pruning." arXiv preprint arXiv:1608.01409 (2016).

---

### Official Review · AnonReviewer1 · 2019-10-23
**Official Blind Review #1**

**Rating:** 3

**Review:**

This paper works on weight quantization in deep networks. The authors propose to utilize the extra state in 2-bit ternary representation to encode large weight values. The authors also propose to use a weighted ridge regularizer which contains a "part of L1" term to make the weights with large values sparse.

The idea is simple and straightforward. However, the paper is not written very well with some typos and some terms defined unclearly. For instance, in the basic quantization method in Section 3.1, 1. What does RELU1 in the first paragraph mean?

Clarification in the experiment section can be further improved. How are the activations for TTQ in section 4.2 quantized? The original TTQ paper also has results for ImageNet, how does the proposed method perform when compared with TTQ on ImageNet?

One major concern is that some popular recent quantization methods are not compared. For instance, [1] also quantized both weights and activations. Can the proposed method outperform it? More comparison with these methods can better illustrate the efficacy of the proposed method.

Another concern is that, though the proposed method has accuracy gain compared with the full-precision baseline and TTQ, the quantization becomes much more complex due to the usage of SLW, does the proposed quantization method cause extra burden to memory access and inference time?

Others:
1. In Tables 3 and 4, "Top-1 Error" => "Top-1 Accuracy"?

[1]. Choi, Jungwook, et al. "Pact: Parameterized clipping activation for quantized neural networks." arXiv preprint arXiv:1805.06085 (2018).

---------- post-rebuttal comments-----------
I really appreciate the authors for their detailed response and additional experiments. However, from the revised manuscript, the proposed method performs significantly worse than some recent quantization methods like PACT and QIL. Thus I keep my rating unchanged.
--------------------------------------------------------


**Experience Assessment:**

I have published one or two papers in this area.

**Review Assessment: Checking Correctness Of Derivations And Theory:**

I assessed the sensibility of the derivations and theory.

**Review Assessment: Checking Correctness Of Experiments:**

I assessed the sensibility of the experiments.

**Review Assessment: Thoroughness In Paper Reading:**

I read the paper at least twice and used my best judgement in assessing the paper.

---

> ### Author Response · Authors · 2019-11-06
> **First Replies**
>
> Thanks to your advice, we can improve the unclear terms. We also agree with your concerns and hope our replies would work for you.
>
> 1) What does RELU1 in the first paragraph mean?
> >> Like your opinion, there are not enough words for ReLU1 in Sec 3.1. The ReLU1 is an activation function, used instead of ReLU activation for quantizing activated values in Basic Quantization (Sec 3.1).
>
> 2) How are the activations for TTQ in section 4.2 quantized?
> >> Actually, the original TTQ paper only quantized weights, not activated values. To compare our method and TTQ in a similar bit-precision, we applied the same activation quantization method as written in the Basic Quantization (Sec 3.1). Taking your advice, we will add more explanations about our TTQ implementation and results made by the original TTQ paper.
>
> 3) How does the proposed method perform when compared with TTQ on ImageNet?
> >> As above, we will add more TTQ results on ImageNet.
>
> 4) More comparison with other state-of-the-art methods
> >> This is a really great question. Actually, we concentrated on the effect of centralized quantization using sparse-large weights. As you mentioned, there are many state-of-the-art quantization methods. To minimize some unexpected or different effects by using state-of-the-art quantization methods such as PACT[1] or QIL[2] when evaluating the centralized quantization, we selected a simple quantization method which is the Basic Quantization (Sec 3.1). Our future work is that applying the centralized quantization to other state-of-the-art quantization methods.
>
> Actually, we tried to put all the contents in 10 pages. It causes not enough explanations. In a reflection of your advice, we will move some sections to appendices and add more detailed comments. And we will add some citations of state-of-the-art quantization methods. Can you recommend a proper section to add the contents of state-of-the-art quantization methods? i) related work, ii) Sec 3.1 (Basic quantization), ii) experimental results (like ABC-net[3]), or iv) multiple choice.
>
> 5) Does the proposed quantization method cause extra burden to memory access and inference time?
> >> That is exact. Using more states in quantization causes an extra burden. Specifically, SLW needs a decoding process at inference-time. To minimize the computational cost of additional bits, we sparsified large weights. In sparse matrix multiplication, multiply operations of convolution layers are skipped when the value of weights is zero. Likewise in our method, almost multiply operations of convolution layers can hold the advantages of ternary weights. We will also add more details about inference to appendices.
>
> We'll update your addressed points. If you want, we can remind you after revising by adding a comment.
>
> If you have more ambiguous terms or any questions, please claim those for revising our paper.
>
> Best Regards.
>
> [1] Choi, Jungwook, et al. "Pact: Parameterized clipping activation for quantized neural networks." arXiv preprint arXiv:1805.06085 (2018).
> [2] Jung, Sangil, et al. "Learning to quantize deep networks by optimizing quantization intervals with task loss." Proceedings of the IEEE Conference on Computer Vision and Pattern Recognition. 2019.
> [3] Lin, Xiaofan, Cong Zhao, and Wei Pan. "Towards accurate binary convolutional neural network." Advances in Neural Information Processing Systems. 2017.

---

### Official Review · AnonReviewer2 · 2019-10-24
**Official Blind Review #2**

**Rating:** 6

**Review:**

This paper is about quantization, and how to represent values as the finite number of states in a low bit width, using discretization. Particularly, they propose an approach to tackle the problems associated with previous ternarization which quantize weights to three values. Their approach is a hybrid weight representation method, which uses a network to output two weight types: ternary weight and sparse-large weights. For the ternary weight, they need 3 states to be stored with 2 bits. The one remaining state is used to indicate the sparse-large weight. They also propose an approach to centralize the weights towards ternary values. Their experiments show that their approach outperforms other compressed modeling approaches, and show an increase in AlexNet performance on the CIFAR-100 while increasing model size by 1.13%.

- Overall, this is an interesting paper, offering a novel solution to tackle the degradation in accuracy occuring in ternary quantization techniques because of the number of quantization steps.  Their method seems technically sound, however I am not familiar with this area, so I would trust more the opinion of other reviewers - subject experts in the matter.
- Their results on AlexNet and ResNet do show an improvement in terms of model accuracy, with only a slight increase in model size. They have also provided extensive experiments studying the benefits of their quantization method, the tradeoff among accuracy and model size.
- I find the abbreviations, being used too often, to be confusing
- The paper is targeted for a very focused audience, and does not give enough background for readers not familiar with "ternarizations".  Even the abstract could benefit from a motivation/ problem statement sentence, as well as less abbreviations being used.

I would vote for acceptance of this paper, although it does seem a too-targeted paper on a specific audience. I would urge the authors to revise writing to make it broader accessible.

**Experience Assessment:**

I do not know much about this area.

**Review Assessment: Checking Correctness Of Derivations And Theory:**

I assessed the sensibility of the derivations and theory.

**Review Assessment: Checking Correctness Of Experiments:**

I assessed the sensibility of the experiments.

**Review Assessment: Thoroughness In Paper Reading:**

I read the paper at least twice and used my best judgement in assessing the paper.

---

> ### Author Response · Authors · 2019-11-06
> **First Replies**
>
> Thanks for your advice!
>
> We also agree that my writing is more focused on the audiences who have some experiences in quantization. Actually, we tried to put all the contents in 10 pages. It causes not enough explanations for the background of quantization. We are going to move some sections (about Sec 3.5 and table 2) to appendices and give more account of the background. After revising, we can remind you by adding a comment if you want.
>
> If you have any questions for understanding other details, please write comments.
>
> Best Regards.

---

### Decision · Program_Chairs · 2019-12-19

**Decision:**

Reject

**Comment:**

The paper proposes a hybrid weighs representation method in deep networks. The authors propose to utilize the extra state in 2-bit ternary representation to encode large weight values. The idea is simple and straightforward. The main concern is on the experimental results. The use of mixed bit width for neural network quantization is not new, but the authors only compare with basic quantization method in the original submission. In the revised version of the paper, the proposed method performs significantly worse than recent quantization methods such as PACT and QIL. Moreover, writing can be improved, and parts of the paper need to be clarified.